# Are Sugar-Reduced and Whole Grain Infant Cereals Sensorially Accepted at Weaning? A Randomized Controlled Cross-Over Trial

**DOI:** 10.3390/nu12061883

**Published:** 2020-06-24

**Authors:** Luis Manuel Sanchez-Siles, Maria Jose Bernal, David Gil, Stefan Bodenstab, Juan Francisco Haro-Vicente, Michelle Klerks, Julio Plaza-Diaz, Ángel Gil

**Affiliations:** 1Research and Nutrition Lab, Hero Group, 30820 Murcia, Spain; mjose.bernal@hero.es (M.J.B.); jfrancisco.haro@hero.es (J.F.H.-V.); michelle.klerks@hero.es (M.K.); 2Institute for Research and Nutrition, Hero Group, 5600 Lenzburg, Switzerland; stefan.bodenstab@hero.ch; 3Pediatric Gastroenterology, Hepatology and Nutrition Unit, Hospital Clínico Universitario Virgen de la Arrixaca, 30120 Murcia, Spain; d.gil.ortega@gmail.com; 4Institute of Nutrition and Food Technology “José Mataix”, Center of Biomedical Research, University of Granada, Avda. del Conocimiento s/n., 18016 Armilla, Granada, Spain; jrplaza@ugr.es (J.P.-D.); agil@ugr.es (Á.G.); 5Department of Biochemistry and Molecular Biology II, School of Pharmacy, University of Granada, 18071 Granada, Spain; 6ibs.GRANADA, Instituto de Investigación Biosanitaria, Complejo Hospitalario Universitario de Granada, 18014 Granada, Spain; 7CIBEROBN (CIBER Physiopathology of Obesity and Nutrition), Instituto de Salud Carlos III, 28029 Madrid, Spain

**Keywords:** cereals, complementary feeding, gastrointestinal tolerance, infancy, infant cereals, sensory acceptability, sugar, sustainable foods, weaning, whole grains

## Abstract

The way infants are fed during the complementary period can have a significant impact on infants’ health and development. Infant cereals play an important role in complementary feeding in many countries. In spite of well documented benefits of a low sugar and high whole grain diet, commercial infant cereals are often refined and contain a high amount of sugars. The aim of the present study was to compare the sensory acceptability, gastrointestinal tolerance and bowel habits of two commercially available infant cereals in Spain with varying sugar and whole grain contents in infants at weaning. Forty-six healthy infants (mean age = 5.2 ± 0.4 months) received one of the two infant cereals containing either 0% whole grain flour and a high sugar content produced by starch hydrolysis (24 g/100 g) (Cereal A) or 50% whole grain flour and a medium-sugar content produced by hydrolysis (12 g/100 g) (Cereal B) in a randomized, triple blind, cross-over controlled trial. Both types of infant cereals were consumed for seven weeks. The cross-over was carried out after seven weeks. Sensory acceptability, anthropometry, gastrointestinal tolerance and adverse events were measured, and results evaluated using a linear regression model. No significant differences were observed between groups in any of the main variables analyzed. Importantly, the long-term health implications of our findings represent a wake-up call for the food industry to reduce or even eliminate simple sugars in infant cereals and for regulatory bodies and professional organizations to recommend whole grain infant cereals.

## 1. Introduction

Complementary feeding is essential for the transition from milk feeding to family foods and plays a crucial role in the life of infants [1]. The way infants are fed during the complementary period can influence taste preferences and eating habits in later child- and adulthood [2,3,4]. Recently, the EAT-Lancet Commission on Healthy Diets from Sustainable Food Systems has published the first scientific targets for a healthy diet from a sustainable food production system that operates within planetary boundaries for food [5]. Two key recommendations included in this report are the reduction of sugar consumption and the promotion of diets with whole grains. Both recommendations are also included in the Sustainable Healthy Diets Guiding principles recently published by the World Health Organization (WHO) and the Food and Agriculture Organization (FAO), which aim to support the efforts of countries to transform food systems to deliver on sustainable healthy diets, contributing to the achievement of the Sustainable Development Goals (SDG) at country level [6].

Most regulatory bodies, health authorities, professional organizations and academia around the world have clearly stated to limit the intake of sugars during infancy [7,8,9,10]. Nevertheless, despite well-documented health benefits of whole grains, there is no clear consensus on the recommendation for the use of this food during infancy and more specifically at the weaning period [11], although some governments and professional organizations (e.g., Spain, Australia and the US) recommend the use of whole grains in infancy and childhood [12,13,14]. However, concerns are noted around potentially high levels of arsenic in brown rice [15].

Infant cereals play an important role in complementary feeding in both developed and low-income countries [16,17,18,19,20]. Infant cereals enhance nutritional intake, compensate iron depletion at the beginning of complementary feeding, foster the development of microbiota and are suitable for a transition from milk to solid foods due to the mild taste and semi-solid texture [11]. Despite these advantages, marketed infant cereals are often refined and contain a high amount of sugars [11,21,22]. The sugars in infant cereals are either added (mostly as sucrose), formed when the starch in the cereals is enzymatically hydrolyzed during processing (producing mainly glucose, maltose and isomaltose) [11] or naturally present. The sugar content naturally present in cereals ranges from 0 to 2.5 g/100 g, depending on the cereal type [23]. Still, it can even reach higher levels such as 30 to 40 g of sugars per 100 g when the cereal is enzymatically hydrolyzed in its formulation [11]. In addition to the sugar content in infant foods, it is also key to look at the amount of whole grains in infant cereals. While the positive impact of whole grain intake on health has been widely studied among adults [24], evidence on infants is mostly lacking [11]. However, it is expected that whole grain intake in infants affects long-term acceptance of whole grain food in later life [11].

A healthy diet for infants is relevant due to its short- and long-term effects on health and eating behavior. Since infant cereals play a crucial role in infant’s food intake, a reduction in sugar content and an increase of whole grains might be beneficial for infant’s health and could help to achieve the recommendations for Sustainable Heathy Diets [5,6]. To the best of our knowledge, no studies are investigating the sensory acceptability and gastrointestinal tolerance of sugar-reduced infant cereals with 50% whole grains at the weaning period. Hence, the purpose of the present study was to compare the sensory acceptability, gastrointestinal tolerance and bowel habits for two commercially available infant cereals with different sugar and whole grain contents in infants between five and eight months of age at the beginning of complementary feeding in Spain. In particular, the paper aims to answer the following questions:1.Is an infant cereal with 50% whole grains and medium-sugar content (12 g/100 g) similarly accepted by infants and parents in comparison to a higher sugar (24 g/100 g) infant cereal without whole grains?2.Is an infant cereal with 50% whole grains well tolerated in infants?

## 2. Materials and Methods

### 2.1. Subjects

Forty-nine healthy infants (mean age: 4.7 ± 0.7 months) were recruited at seven healthcare centers that were allocated to Health Area I of “Hospital Universitario Virgen de la Arrixaca” in Murcia (Spain). The eligible infants had a gestational age of 37–42 weeks, a birth weight higher than 2500 g and were between the age of 4–5 months at the start of the study. At time of study enrolment, they were not yet introduced to complementary foods were exclusively formula-fed at least from four months of age and did not use antibiotics at least 15 days before the enrolment. Infants were excluded when they had congenital diseases at birth, acute conditions at recruitment, diseases that affect metabolism or intake of food (including food allergies and intolerances). Infants that were participating in other studies were not allowed to participate.

Pediatricians from the primary care health centers who collaborated in this study were the ones who were assigned by law to the infants of this study. Those pediatricians did all periodic revisions and monitored any possible adverse events (AEs). An AEs alert system was specially designed to detect not only the AEs usually related to complementary feeding (intolerances, allergies …) but any other sign or symptom of discomfort in children.

The study was conducted according to the guidelines established in the Declaration of Helsinki (2013) [25] as well as the International Conference on Harmonization Guideline for Good Clinical Practice. The protocol was approved by the Ethics Committee of “Hospital Universitario Virgen de la Arrixaca” (Murcia, Spain) and written informed parental consent was obtained for each infant before inclusion. The study was registered in the ClinicalTrials.gov as NCT02781298.

### 2.2. Products

The two infant cereals used in this experiment were in full compliance with European Directives and Regulations [26,27], and commercially available products from Hero España S.A (Murcia, Spain). Incorporation of infant cereals into the children’s diet was upon pediatrician’s recommendation, which in our study occurred at a mean age of 5.2 ± 0.4 months. This is in line with the European recommendations for complementary feeding [1,8]. The mode of preparation (i.e., quantity of cereals prepared with formula milk served in bottle or plate) and the moment and frequency of consumption were decided by the parents in consultation with the pediatricians.

One of the two infant cereals contained 100% refined cereals and a high sugar content produced by starch hydrolysis (24 g/100 g) (Cereal A) and the other cereals studied had 50% of whole grain flour and a medium sugar content produced by hydrolysis (12 g/100 g) (Cereal B). The ingredients of Cereal A were: Hydrolyzed cereal flours (wheat, corn, rice, oat, barley, rye, sorghum and millet), minerals, natural flavor and vitamins. The ingredients of Cereal B were: Partially hydrolyzed cereal flours (wheat, whole grain wheat (50%), corn, rice, oat, barley, rye, sorghum and millet), minerals, natural flavor and vitamins. The nutritional composition of the two infant cereals used in this study is depicted in Table 1. Both infant cereals were labeled equally and were provided, designed, produced and coded by Hero España S.A.

### 2.3. Study Design

The study had a randomized, triple blind, cross-over and controlled prospective design and was carried out from September 2015 to September 2016. The total duration of the intervention was 14 weeks. Both types of infant cereals were consumed for 7 weeks. The cross-over was carried out after 7 weeks. Randomization of the participants was performed with a free internet application (www.randomization.com). The randomization schedule was in blocks of 4 for each study site. The investigators, healthcare providers/personnel, data analysts and the infants’ parents were blinded to the group allocation. The blinding was broken once the statistical analysis was completed.

### 2.4. Data Collection and Study Outcomes

An overview of the data collection points for each of the variables is shown in Figure 1. Infants in feeding sequence AB received first the Cereal A, followed by the Cereal B, whereas the infants in sequence BA received first the Cereal B followed by the Cereal A. During the 14-week intervention there were 7 visits. At visit 1 baseline anthropometric measurements and stool samples were taken. After the fourth visit the participants changed intervention (i.e., product). The subjects were visited three times during each intervention period. The average ages of subjects in each point of visit was: Visit 2 = 5.4 months, visit 3 = 6.1 months, visit 4 = 6.9 months, visit 5 = 7.2 months, visit 6 = 7.9 months and visit 7 = 8.7 months. Sensory acceptability was evaluated at visits 2, 3, 5 and 6 and anthropometric measurements were evaluated at visits 1, 4 and 7. Furthermore, regurgitations, gassiness, frequency of defecation, and stool consistency were evaluated at visits 2, 4, 5 and 7. Adverse events were monitored throughout the study.

The general data were collected with two types of questionnaires. Pediatrician’s questionnaires included infant characteristics, anthropometric measurements and adverse events. Parents’ questionnaires included the sensory test, variables of gastrointestinal tolerance and stool characteristics.

To study the possible effect in the intestinal microbiota of the intake of different infant cereals we collected stool samples at visit 1 (baseline) and at visit 4 (before the cross-over of cereals). These stool samples were collected from nappies in a plastic feces container, frozen immediately after collection by the parents at home and stored until they were taken to a central collection site. Feces were then kept at −80 °C until analysis. The specific methodology and the results obtained for the fecal microbiota will be published in a separated paper.

#### 2.4.1. Sensory Acceptability

Sensory acceptability of the infant cereals was measured in both infants and parents, where parents interpreted the “infant’s reaction” before reporting their own. The infant’s acceptability was assessed via two variables: The “infant’s reaction” towards the infant cereals and the “ingested amount”. The infant’s reaction was measured by means of a 4-point hedonic validated scale [29,30,31]. This scale uses the following scores, “0 = very negative” if the infant spits out the food, frowned, pushed the spoon away or stopped eating; “1 = negative” if the infant ate a couple of spoonful, grimaced and stopped eating; “2 = positive” if the infant ate some of the food without a specific reaction; “3 = very positive” if the infant accepted the first spoonful immediately and displayed signs of contentment, such as a relaxed face or a smile. All scores on the scale were accompanied with a corresponding smiley-face to guide the parents. The ingested amount was measured via a 6-point scale with scores ranging from “0 = he/she ate nothing” to “5 = he/she ate everything”. Subsequent to scoring the infant’s acceptability, parents rated their own overall impression and liking of taste on a 7-point hedonic scale ranging from “1 = dislikes very much” to “7 = likes very much.”

#### 2.4.2. Other Secondary Outcomes

##### Anthropometric Measurements

Anthropometric measurements included body weight, length and head circumference. These measurements were performed in duplicate by a pediatrician. Infants’ body weight was measured in an electronic scale (accuracy ± 10 g). Length of the infants was measured via a stadiometer. When the second measurement deviated more than 0.4 cm, the measurement was performed a third time. Finally, the head circumference was measured at the height of the occipital frontal. When the second measurement deviated more than 0.2 cm, the measurement was performed a third time.

##### Gastrointestinal Tolerance

Gastrointestinal tolerance included the frequency of regurgitation and gassiness. Data was collected during the last 48 h before visits 2, 4, 5 and 7 by the parents. Frequency of regurgitation was measured on a 3-point scale, 0 = no episodes, 1 = one episode, 2 = two or more episodes [32]. Gassiness was determined on a 4-point scale, 0 = no episodes, 1 = low episodes, 2 = moderate episodes, 3 = excessive episodes [33].

##### Bowel Habits

Bowel habits were determined by the frequency of defecation and consistency of the infant’s stool. Data was collected during 48 h before visits 2, 4, 5 and 7. The frequency of defecation was measured as number of depositions per day. The stool consistency was evaluated according to the methodology of on a 5-point scale, 1 = watery, 2 = loose, 3 = soft, 4 = formed, 5 = hard [34].

##### Adverse Events

Although no adverse events (AEs) derived from a common and habitual practice was expected, any possible AE was monitored. Indeed, an AEs notification system, similar to that of a clinical trial with drugs, was designed following international recommendations [35]. The AEs alert system was specially designed to detect not only the AE usually related to complementary feeding (intolerances, allergies…) but any other sign or symptom of discomfort in children. All reported symptoms were considered AEs, even though they were not related to the intervention. AEs were classified as mild, moderate and severe, and according to their possible relationship with complementary feeding as unrelated, possibly related and related to the intervention. Any serious AEs or in possible relation to the intervention detected by the pediatrician had to be reported immediately to the principal investigator. The AEs were reported as the number and type of infections/symptoms and medical treatments.

The medical occurrences were clustered in seven groups of infections/symptoms (0 = no episodes, 1 = upper respiratory tract infections including otitis, 2 = lower respiratory tract infections, 3 = gastrointestinal infections including parasitosis, 4 = other infections like varicella, 5 = dermatological complications, 6 = non-infectious gastrointestinal symptoms like constipation or vomiting, 7 = other complications) and seven groups of medical treatments (0 = no treatments, 1 = antihistamines, anti-inflammatory and bronchodilators, 2 = antipyretics and analgesics, 3 = antibiotics and anti-infectives, 4 = probiotics, 5 = laxatives, 6 = inhaled corticosteroids or topical, 7 = emollient and other topical creams).

### 2.5. Data Analysis

Sample size was calculated based on the primary outcome (i.e., parent’s perception of “infant’s reaction”) using a standard deviation of 0.5 and detecting a 0.3 reduction in the response. With a power of 80% and a significance level of *p* < 0.05 (two-sided test), the minimal required sample size was 22 infants per treatment group [29,36].

Baseline characteristics (i.e., gender, birthweight, age, current weight, current length and head circumference) were described in both groups of feeding sequences (AB vs. BA). Continuous variables are reported as mean ± standard deviation (SD) and categorical variables are reported as percentages (%). To detect differences between the feeding groups in each variable an unpaired student’s T test and Pearson chi-square test were used for variables expressed continuous or as percentages, respectively.

Main outcomes: We looked at eight outcome variables divided in four groups: The degree of liking or acceptability in infants (variables “infant’s reaction” and “ingested amount”), degree of liking or acceptability in parents (variables “overall impression” and “taste”), gastrointestinal tolerance (variables “regurgitations” and “gases”) and bowel habits (variables “defecations per day” and “stool consistency”). All variables were treated as continuous with the scales described in Section 2.4 as described above. The only exception is that the stool consistency was simplified in a 4-category score because the two first categories “watery” and “loose” were collated into one. The analysis of each of the eight outcome variables was done separately with a linear regression model. We designed a linear regression model that estimates the outcome mean at four time moments: T1 (visit 2), T2 (visit 3), T3 (visit 5) and T4 (visit 6) that are, respectively, just after beginning of first cereal, end of first cereal, just after beginning of second cereal and at the end of second cereal. We compared between arms the estimated outcome means at T1 (initial reactions to first cereal). Then, by using linear combinations of the parameters, we calculated the mean outcome changes between times: T2–T1 captured how the infant gets used to the first cereals, T3–T2 captured the effect of switching cereals and T4–T3 captured how the infant gets used to the second cereals. We also compared these changes between the two groups. Each model included a random effects term by individual to account for repeated measures.

Adverse events and medications: The number of visits and patients that reported each adverse event when taking each cereal formula was computed, and proportions were compared between cereals using Fisher’s exact tests. The same procedure was followed to analyze proportion of treatments given to children taking each kind of cereal.

All results with a *p*-value < 0.05 were considered statistically significant. Statistical analyses of the data were performed using R-studio (version 3.5.1) (R-studio, Inc., Boston, MA, USA).

## 3. Results

### 3.1. Subjects Characteristics or Characteristics of Study Population

A total of 49 infants were eligible for inclusion, one subject was declined and 48 were included in the study (feeding sequence AB: *n* = 22; feeding sequence BA: *n* = 26). Forty-six subjects completed the study, 22 in feeding sequence A and 24 in feeding sequence B (drop-out rate of 8%) (Figure 2).

There were no differences between the two feeding sequences in terms of gender, weight of birth, age, weight, length and head circumference at baseline before treatment (Table 2).

### 3.2. Sensory Acceptability

The statistical results obtained in the sensory tests are shown in Table 3.


**Infant’s reaction**


At T1 (initiation of first infant cereals) group AB was taking Cereal A and their average degree of liking was 2.36 while group BA that was taking Cereal B had an average of 2.40 without significant differences (*p* = 0.8178). Between T2 and T3, when the infant cereals are changed, there was no statistical evidence that changes are different between the groups (*p* = 0.4550) (Table 3). Those going from Cereal A to Cereal B had a small mean change of −0.04, while those going from Cereal B to Cereal A had a small mean increase of 0.08, but there was not a significant difference in infant’s reaction between Cereal A vs. Cereal B (*p* = 0.4550).

Both infant cereals (Cereal A and Cereal B) were highly accepted with mean scores higher than 2 in all the visits (Figure 3).


**Ingested amount**


Figure 4 shows that infants in feeding sequence AB had a higher intake than the infants in feeding sequence BA throughout the study, although we observed no significant differences between feeding groups. The mean scores were higher than 4 in all visits.

The infant cereals consumed (Cereal A vs. Cereal B) neither influenced the ingested amount. The infants in group AB tended to ingest more as their average intake was 4.82 while group BA had an average of 4.17, with a significant difference between the two means (*p* = 0.0082). During the first period, the AB group tended to decrease its average intake while the BA group tended to increase it although not significantly (*p* = 0.1199). Between intervention change visits no significant differences were observed (*p*= 0.4851) (Table 3).


**Parent’s acceptability**



**Parent’s taste**


Parent’s average taste was very similar when they started their first cereal, and changed little over the first period. However, when they switched cereals, those going to Cereal B seemed to increase their average taste (0.18). In contrast, the average taste of those moving to Cereal A seemed to decrease (−0.33), with some statistical evidence for the difference between these trends (*p*-value = 0.06). However, during the second period the parents giving Cereal A increased their taste levels catching up with the average taste in the Cereal B group that almost did not change. The difference in trend between groups was statistically significant (*p* = 0.04).


**Overall impression**


The parent’s overall impression at the beginning of the first cereal was similar in both groups being slightly higher in the group of AB than in BA (4.86 vs. 4.46, respectively). However, as shown in Table 3 the average of overall impression of both groups became more similar in every visit, and none of the mean changes or differences between groups were statistically significant.

### 3.3. Other Secondary Outcomes

#### 3.3.1. Anthropometric Measurements

During the intervention period, all infants had a normal growth pattern, since mean values of weight, length and head circumference for both intervention groups registered in each visit were not significantly different to the standard curves reported by WHO (WHO, 2006); (results not shown).

#### 3.3.2. Gastrointestinal Tolerance

Regurgitation: at the beginning of the first cereal, infants on Cereal A showed higher average episodes of regurgitation than the infants taking Cereal B (1.64 vs. 1.21, respectively), although the differences were not statistically significant (*p* = 0.2518). However, during the first period average regurgitation decreased significantly in infants taking Cereal A (−0.91) and hardly changed in those taking Cereal B (−0.08), being the difference of trends statistically significant (*p* = 0.0245). When infants switched cereals they tended to show a small (non-significant) increase in regurgitations but then they reduced again their average as they get used to the cereal in the second period (Table 4).

Gassiness: when starting the first cereal both groups had similar episodes of gassiness with no significant differences between them (1.95 vs. 2.10, *p* = 0.6947). In both groups there was also a decrease of events of gassiness at the end of the first period (−0.50, −0.56, respectively). After cross-over, the intake of another cereal did not lead to differences in gassiness (*p* = 0.7971). Both groups followed parallel trends with reduction of gassiness in both periods as the infant got used to the formula with a small rebound when switching the cereals. There seems to be no significant difference between cereals (Table 4).

In general, the tolerance seemed to improve over time in both diets.

#### 3.3.3. Bowel Habits

Frequency of depositions started at about the same level in both groups and stayed stable across both periods, among 1.4 and 1.7 depositions per day. There were no significant trends over time in both groups and no significant differences between the cereals studied (*p* = 0.6089) (Table 5).

The average stool consistency started at similar levels in both groups around 2.38 (Table 5). There was a quick increase in stool consistency in both groups in the first period but no significant difference between the trends of the groups were observed (*p* = 0.2682). From that point onward there were no statistically significant changes in any of the groups (*p* = 0.3940, *p* = 0.9199) (Table 5). We observed a physiological increase in stool consistency, being softer at the beginning and more formed at the end, when the age of the infant was higher (Figure 5).

#### 3.3.4. Adverse Events

The children in our study did not suffer any pain and did not find any difficulties with the incorporation of infant cereals in their diet. The infant cereals were introduced at an adequate age in line with the recommendations of the pediatricians and the European recommendations for complementary feeding. Throughout the whole study period the children were controlled for any issues by their pediatricians. Of the 46 subjects, no AEs were reported during the two study periods (seven + seven weeks) for 22 subjects (48%). AEs were reported for 24 subjects. In total, 24 mild AEs were reported among 17 children when taking Cereal A, and another 24 mild AEs were reported in 14 children while taking Cereal B. Of these, 77% were respiratory tract infections (58.3% were upper respiratory tract infections including otitis and 18.7% were lower respiratory tract infections); 12.5% were gastrointestinal infections including parasitosis and 2.1% were non-infectious gastrointestinal symptoms, like constipation or vomiting; 2.1% were other infections like varicella and 6.3% were dermatological complications. None of the AEs that occurred were related to the intake of the infant cereals or other foods. Ten children received 15 treatments while taking the Cereal A and 13 patients received 21 treatments while taking Cereal B. Out of all treatments, 55.6% were antihistamines, anti-inflammatory and bronchodilators, 19.4% antipyretics and analgesics, 8.6% antibiotics and anti-infectives, 2.8% probiotics, 2.8% laxatives, 8.3% were inhaled corticosteroids or topical and 2.8% were emollient and other topical creams. A statistical difference was found between both periods of the study, with 34 infections in the first period vs. 14 infections in second period (*p* = 0.04). The incorporation of the children to the kindergarten and/or the autumn season could be a reason for the higher rate of infections during the first period. Additionally, a difference was found between the number of treatments used in both periods—28 in the first period vs. 8 in the second period (*p* = 0.007). However, there were no significant differences in the incidence of AEs, type of adverse events and treatments applied between when using one infant cereal or the other.

## 4. Discussion

This study has investigated the sensory acceptability of two commercially available infant cereals, varying in sugar and whole grain content, by means of a 14-week randomized cross-over controlled trial in infants and their parents in Spain. Importantly, both infant cereals included in this study were well accepted by the infants over the whole course of the intervention, yet no statistical difference was found between them, neither for the first intake of the cereals nor when the cereals were switched. Additionally, both infant cereals were equally well accepted by the parents. Measurements on gastrointestinal tolerance of the infants were also included in this research and indicated a significant higher percentage of infants with regurgitations for the high sugar cereals (Cereal A) at the second and third visit. No significant differences were found in the remaining variables analyzed between the groups.

Infant cereals acceptability studies are scarce. To our knowledge, this is the first study assessing infant cereals acceptability and gastrointestinal tolerance in infants over time. Previously, Haro-Vicente et al., (2017) [29] investigated the sensory acceptability of infant cereals with a varying whole grain content (30% vs. 0%) in an eight-day trial indicating that the cereals were equally accepted by both infants and parents [29]. Despite the higher whole grain percentage difference in our study (50% vs. 0%) and additional varying sugar content (12 g/100 g vs. 24 g/100 g), sensory acceptability still did not differ between the two groups for both cereals. Remarkably, even when infants had refined infant cereals with a high sugar content for seven weeks (first period of intervention), we have shown that intake of infant cereals with 50% whole grains and a medium-sugar content for the following seven weeks is still highly and similarly accepted.

Although infants are born with an innate preference for sweet taste, they are capable of learning to like a variety of new flavors and foods during the complementary feeding period [37], regardless the level of sweetness. Interestingly, the fact that we did not find any difference in infant or parent acceptability with infant cereals differing in sugar content from 24% vs. 12% is apparently in contradiction with the general belief that infants prefer sweet tasting foods [38,39]. Two reasons might explain this issue: (1) There is a small difference in sugar content per portion. Per portion (±20 g) the sugar content in the infant cereals only differ approximately 2.5 g; and (2) infant cereals are usually prepared with human milk or infant formula. We could hypothesize that their sweet taste might mask the taste of the infant cereals [29]. Breast milk contains about 7 g of lactose/dL [40] and infant formulas, when diluted, provide a similar level of lactose. However, to our knowledge, the direct correlation between lactose content and sweetness in breast milk or infant formula has not been studied in-depth yet. Only a recent study using a sensory panel indicated that breast milk is perceived as sweet and its sweetness correlated with the carbohydrate content in the milk [41]. The preferred degree of sweetness for infants remains unknown. In newborns, a concentration of 0.73 M sucrose (25 g/100 mL) elicits positive responses, like smiling, sucking movements and facial relaxation [42,43]. This was not compared with other sucrose concentrations and different types of free sugars (i.e., glucose, maltose and lactose). Moreover, it is crucial to take into consideration that sweetness perception does not only depend on the entire sugar content, but also on genetic traits [44] and food matrix [45].

The consumption of sugar is country dependent and reformulation strategies to reduce added sugar content only have significant effects if the target food represents an important contribution to the population’s diet [46]. In that sense, the intake range of infant cereals in Spain is roughly 10–40 g/day from four to 12 months [22]. If we were to apply sugar reduction as indicated in our study, the daily intake of sugar from infant cereals would be lowered from approximately 2.4–9.6 to 1.2–4.8 g, which would significantly contribute to meeting the objective of reduction of free sugars to <5% of daily energy intake in this sensitive population [7,9].

In addition to the different sugar content, the infant cereals used in the present study had two different whole grain contents—0% vs. 50%. A broad consensus exists about the health benefits that come with eating whole grain in adults, such as reduced levels in blood cholesterol, body weight and non-communicable diseases [24]. Although a consistent recommendation about whole grain intake is lacking, it is generally acknowledged that whole grain intake is preferred over refined grain intake [11]. Like in adults, whole grain recommendations for infants are lacking or inconsistent; however, it is expected that the inclusion of whole grain in the infant’s diet might be promising for the future [11]. Whole grain intake could be a major factor to increase gut health due to the complex carbohydrates in whole grain cereals (dietary fiber, resistant starch, oligosaccharides) that could shape digestion and absorption in the small intestine and, also, as they are fermented in the large intestine they are working as a prebiotic [47]. Childhood constipation might be relieved by the increase in whole grain intake; however, randomized controlled trials are lacking to make firm recommendations [48].

The present study, designed as a cross-over randomized controlled trial, gave us the opportunity to compare bowel habits after consuming whole grain infant cereals as compared to refined cereals infants. In this study, we used the frequency of defecation and stool consistency as markers for bowel habits. Both parameters can be influenced by age and feeding practices of the infant [49]. This was the case for stool consistency; however, no difference in the frequency of defecations was found between the products nor over time. The bowel habit is a useful marker of intestinal function, especially in the colon. A higher frequency of deposition and softer and watery stools have been shown in breastfed infants compared with formula-fed infants; however, these differences disappear after the weaning period [49]. Intake of whole grain cereals, as compared to refined cereals, could be an important factor in these markers due to the complex carbohydrates in whole grain cereals [47]. In fact, although there are no studies about the influence of feeding profile on stool patterns in infant and toddlers during complementary feeding, studies in older children and adults exist. For instance, Grasten et al., (2000) [50] compared bowel function in middle-aged women and men. Whole-meal rye bread significantly increased fecal output and fecal frequency and shortened mean intestinal transit time when compared with wheat bread. The intake of whole grains could improve constipation in childhood due to the increase of dietary fiber and other bioactive compounds that could also help. Well-designed additional randomized controlled trials are necessary to make a formal recommendation in this regard.

Our results did not provide evidence for major differences in the number and type of adverse effects depending on the type of cereal consumed. A possible issue that has been associated with the intake of whole grain cereals is the higher phytate content vs. the refined counterpart and its effect on bioavailability of calcium, iron and zinc. Values higher than 0.24 for the molar phytate/calcium ratio seems to compromise the availability of calcium [51] and the molar phytate/iron ratio should be lower than 0.4 to achieve adequate iron absorption [52]. In addition, values of this ratio above 12–15 compromise the availability of zinc [51,53]. Nevertheless, the processing of infant cereals contributes to decreasing the phytate content [28]. Significant decreases occurred in the mixtures of wheat, barley and rye with losses around 50% of their initial content, while minor changes were observed for sorghum and millet flours [54].

The estimated phytate content of cereals in our study were: Cereal A (143.5 mg/100 g, equivalent to 0.23 mmol/100 g) and Cereal B (176.8 mg/100 g, equal to 0.28 mmol/100 g) and the phytate/mineral molar ratios for Cereal A and B, respectively, were 0.05 and 0.06 for calcium; 1.40 and 1.75 for iron; and 9.8 and 12.2 for zinc. However, it must be considered that infant cereals are always reconstituted with breastmilk or infant formula prior to consumption, decreasing the phytate/mineral molar ratios. Anyhow, prior evidence points out that decreasing phytate beyond current contents in cereals had no effect on growth, development or incidence of diarrheal or respiratory infections [55] and had little long-term effect on the iron and zinc status of Swedish infants [56].

There may be some limitations in the present study. First, both the sugar and whole grain content differed between the infant cereals. Therefore, our findings could be attributed to either one of those factors or to their combined effect. Nevertheless, there is prior evidence indicating the sensory acceptability of infant cereals with 30% whole grains with no differences vs. the refined counterpart [29]. Furthermore, as both lower sweetness and addition of whole grains may negatively impact the sensory properties of infant cereals, our results are even more interesting as the whole grain/medium-sugar infant cereals were similarly accepted. Second, as the infants grow older, complementary feeding was added to their diet. It is unclear if this might have influenced the sensory acceptability or gastrointestinal reaction of the infants. Moreover it needs to be taken into consideration that the estimation of infant acceptability by parents may reflect their own personal preferences and they might use different criteria to assess liking of their children [57,58]. The advantage of this 4-point scale is that at each scale point a small description is given about infant behavior, which might reduce this bias. Prior research using this scale did not detect any differences between infant cereals [29], but seemed to be more effective in comparing different food groups [30,31]. Third, in our study, parent liking of the infant cereals was moderately positive for both cereals, but no significant difference was found between them. This is of interest, since the role of parents in infant food behavior is crucial [59]. In our study, parents evaluated the two infant cereal samples blind; however, this does not reflect normal consumer behavior. Previous research indicates that opinions about the social responsibility of the brand (e.g., regarding the environment), convenience and price and safety are important factors in parent brand choice for infant foods [60].


**Implications for food manufacturers and policymakers**


Several implications for *the food industry, policymakers, pediatricians and parents* can be derived from our findings. First and foremost, sugar can be reduced or even eliminated and refined grains can be substituted by whole grains in infant cereals without compromising on taste. This would result in healthier, more sustainable and more natural (less processed) cereals. Such products would not only fit better with international health agencies and pediatric associations (e.g., FAO, WHO, The European Society for Paediatric Gastroenterology Hepatology and Nutrition (ESPGHAN), Scientific Advisory Committee on Nutrition (SACN), The American Academy of Pediatrics (AAP), among others), but would also be in line with sustainable recommendations from the SDGs and the Planetary Health Diet as well as recent food international consumer trends [61] and measurements on food naturalness [62]. Overall, given our evidence that less sugar and a higher level of whole grain in cereals do not compromise its acceptability, policy makers and health care organizations can explicitly encourage infant food manufacturers in their new product development or product reformulations to reduce sugar levels and increase whole grain content in infant cereals.

## 5. Conclusions

This 14-week Randomized Controlled Trial (RCT) investigated the sensory acceptability, gastrointestinal tolerance and bowel habits in infants consuming two infant cereals with varying whole grain and sugar contents. Infant cereals play a prime role in complementary feeding, which in turn has significant consequences on infant’s health and future development. Importantly, our main finding was that infant’s and parent’s acceptability did not significantly differ between tested infant cereals. In addition, no major differences in gastrointestinal tolerance and bowel habits were found. We believe our research considerably adds to the literature, given the evident scarcity in RCTs and sensory acceptability studies in infant foods. Overall, our findings represent an opportunity for the food industry to reduce or even eliminate sugars and increase the levels of whole grains in in infant cereals. This would be consistent to current health and sustainability recommendations from the EAT-Lancet Commission and international health and pediatric organizations.

## Figures and Tables

**Figure 1 nutrients-12-01883-f001:**
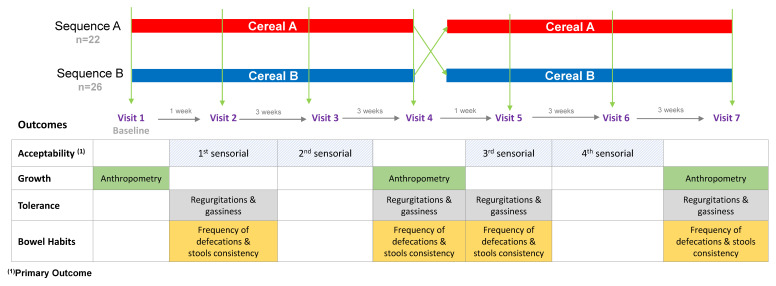
Diagram of the study protocol: Intervention, time-points and measured variables. Cereal A—0% whole grain and 24 g/100 g and Cereal B—50% whole grain and 12 g/100 g sugars.

**Figure 2 nutrients-12-01883-f002:**
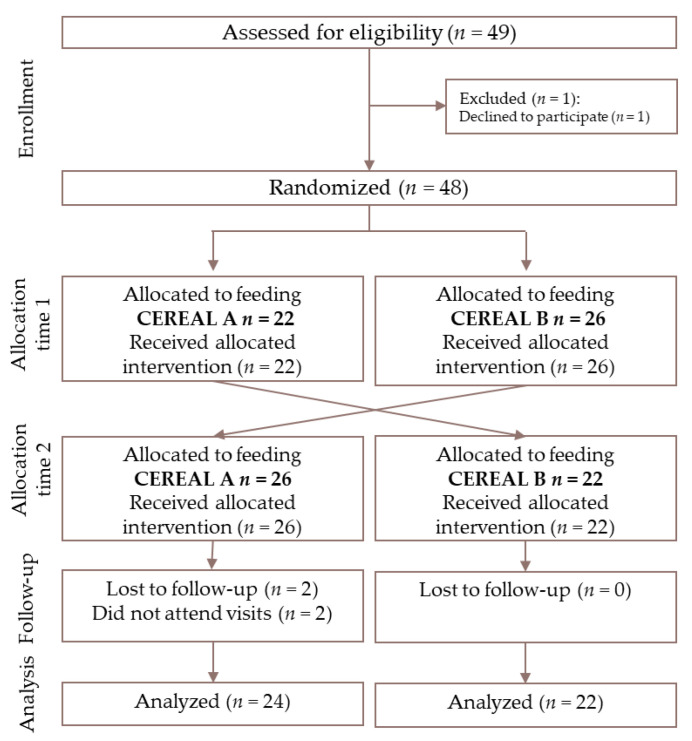
Consort flow chart adapted to cross-over design. Cereal A—0% whole grain and 24 g/100 g sugar; Cereal B—50% whole grain and 12 g/100 g sugars.

**Figure 3 nutrients-12-01883-f003:**
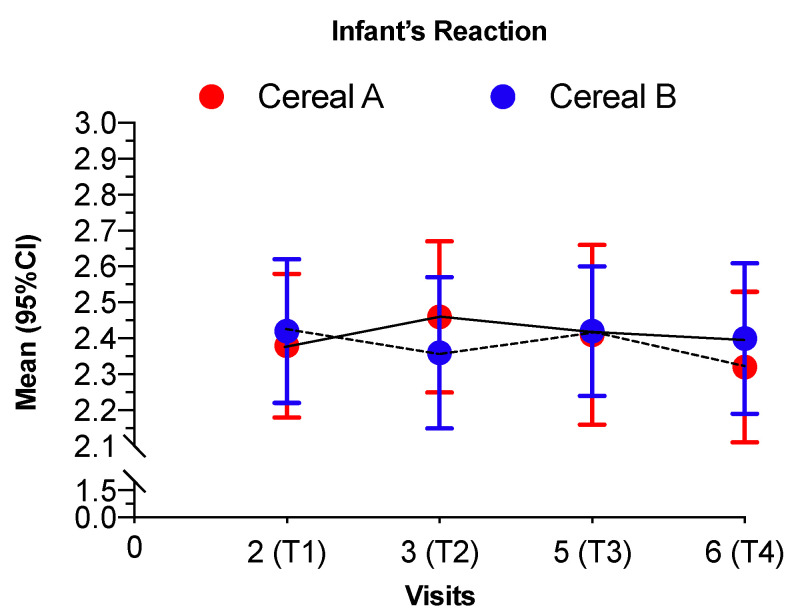
Scores for infant’s reaction in each visit for both intervention groups. Scale—0 = very negative, 1 = negative, 2 = positive, 3 = very positive. Values are shown as mean and 95% confidence intervals. Continuous line represents the intervention group AB and the discontinuous line represents the intervention group BA. Cereal A—0% whole grain and 24 g/100 g sugar; Cereal B—50% whole grain and 12 g/100 g sugars.

**Figure 4 nutrients-12-01883-f004:**
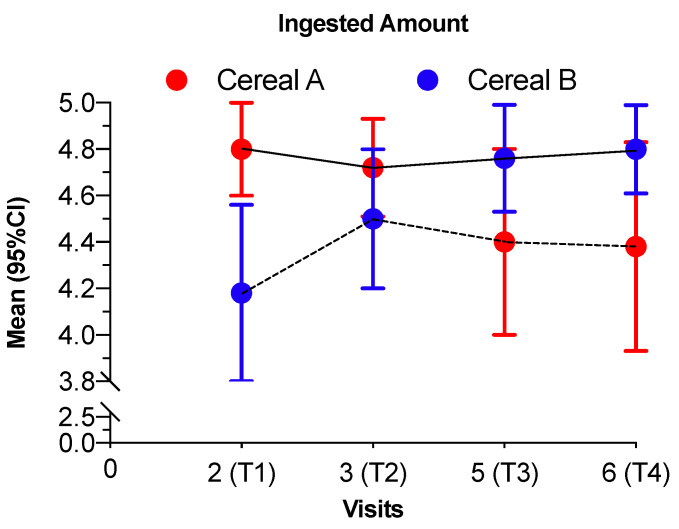
Scores for ingested amount in each visit for both intervention groups. Scale—0 = he/she ate nothing to 5 = he/she ate everything. Values are shown as mean and 95% confidence intervals. Continuous line represents the intervention group AB and the discontinuous line represents the intervention group BA. Cereal A—0% whole grain and 24 g/100 g sugar; Cereal B—50% whole grain and 12 g/100 g sugars.

**Figure 5 nutrients-12-01883-f005:**
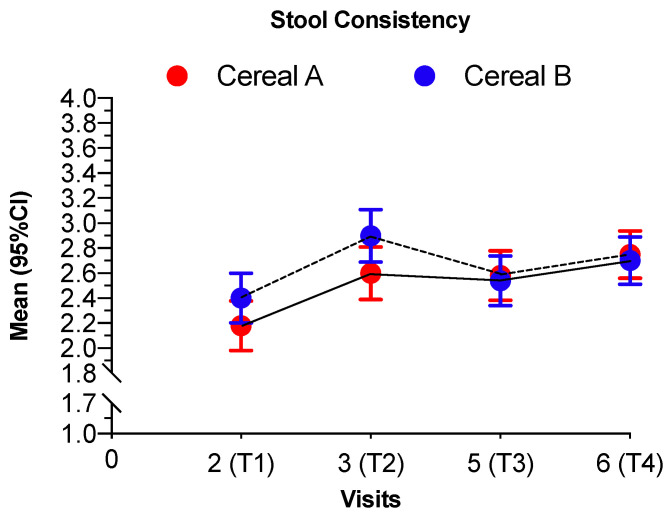
Stool consistency of the infants in each visit for both intervention groups. Scale—1 = watery, 2 = soft, 3 = formed, 4 = hard. Values are shown as mean and 95% confidence intervals. Continuous line represents the intervention group AB and the discontinuous line represents the intervention group BA. Cereal A—0% whole grain and 24 g/100 g sugar; Cereal B—50% whole grain and 12 g/100 g sugars.

**Table 1 nutrients-12-01883-t001:** Nutritional composition of the two infant cereals used in the study (Cereal A—0% whole grains and 24 g/100 g sugars and Cereal B—50% whole grains and 12 g/100 g sugars).

Nutrients (per 100 g)	Cereal A	Cereal B
Energy (kcal)	376	375
Protein (g)	9.85	10.13
Carbohydrates (g)	79.50	75.05
Sugars (g)	24.10	12.07
Fat (g)	1.33	2.02
Fiber (g)	4.03	7.21
Calcium (mg)	160	160
Iron (mg)	6.23	8.16
Zinc (mg)	0.63	1.09
Phytate (mg) *	143.51	176.83

* Data obtained from similar products and processing from Frontela (2007) [28].

**Table 2 nutrients-12-01883-t002:** Subjects characteristics at birth and enrolment.

Variable	Feeding Sequence		*p* Value
	AB (*n* = 22)	BA (*n* = 24)	
Gender infant, *n* (%)			0.777
Male	11 (50)	11 (46)	
Female	11 (50)	13 (54)	
Birth weight (kg) (mean ± SD)	3.27± 0.33	3.26 ± 0.45	0.905
**Baseline characteristics**			
Age (months) (mean ± SD)	5.17 ± 0.38	5.24 ± 0.42	0.564
Weight (kg) (mean ± SD)	7.41 ± 0.87	7.36 ± 0.78	0.835
Length (cm) (mean ± SD)	64.50 ± 2.78	65.17 ± 2.51	0.397
Head circumference (cm) (mean ± SD)	42.28 ±1.51	42.61 ± 1.36	0.441

**Table 3 nutrients-12-01883-t003:** Changes in variables for sensory acceptability between visits (infant’s reaction, food intake, parent’s taste and parent’s overall impression). T1: Initiation of first formula, T2: Follow-up of first formula, T3: Initiation of second formula, T4: Follow-up of second formula.

Outcome of Sensory Acceptability	Visit	Group AB	Group BA	*p*-Value
Infant’s reaction	at T1	2.36 (2.13, 2.59)	2.40 (2.18, 2.62)	0.8178
	T2–T1	0.13 (−0.11, 0.37)	−0.03 (−0.25, 0.20)	0.3461
	T3–T2	−0.04 (−0.28, 0.20)	0.08 (−0.14, 0.31)	0.4550
	T4–T3	−0.05 (−0.29, 0.19)	−0.13 (−0.35, 0.10)	0.6580
Ingested amount	at T1	4.82 (4.47, 5.16)	4.17 (3.84, 4.50)	0.0082
	T2–T1	−0.09 (−0.51, 0.33)	0.38 (−0.03, 0.78)	0.1199
	T3–T2	−0.00 (−0.42, 0.42)	−0.21 (−0.61, 0.20)	0.4851
	T4–T3	0.13 (−0.30, 0.57)	−0.04 (−0.45, 0.36)	0.5600
Parent’s taste	at T1	4.73 (4.22, 5.24)	4.81 (4.32, 5.30)	0.8217
	T2–T1	0.00 (−0.39, 0.39)	−0.06 (−0.44, 0.32)	0.8319
	T3–T2	0.18 (−0.21, 0.57)	−0.33 (−0.71, 0.04)	0.0642
	T4–T3	0.00 (−0.40, 0.41)	0.58 (0.21, 0.96)	0.0407
Parent’s overall impression	at T1	4.86 (4.34, 5.39)	4.46 (3.96, 4.96)	0.2764
	T2–T1	−0.09 (−0.49, 0.31)	0.13 (−0.26, 0.51)	0.4484
	T3–T2	−0.14 (−0.54, 0.27)	−0.00 (−0.38, 0.38)	0.6318
	T4–T3	0.01 (−0.40, 0.43)	0.08 (−0.30, 0.47)	0.8110

**Table 4 nutrients-12-01883-t004:** Changes in regurgitation and gassiness between visits. T1: Initiation of first formula, T2: Follow-up of first formula, T3: Initiation of second formula, T4: Follow-up of second formula.

Outcome var.	Time	Group AB	Group BA	*p*-Value
**Regurgitation**	at T1	1.64 (1.11, 2.16)	1.21 (0.71, 1.71)	0.2518
	T2–T1	−0.91 (−1.43, −0.39)	−0.08 (−0.58, 0.41)	0.0245
	T3–T2	0.14 (−0.38, 0.65)	0.06 (−0.43, 0.56)	0.8397
	T4–T3	−0.27 (−0.79, 0.24)	−0.35 (−0.85, 0.14)	0.8235
**Gassiness**	at T1	1.95 (1.42, 2.49)	2.10 (1.59, 2.62)	0.6947
	T2–T1	−0.50 (−0.96, −0.04)	−0.56 (−100, −0.12)	0.8480
	T3–T2	0.09 (−0.37, 0.55)	0.19 (−0.25, 0.63)	0.7671
	T4–T3	−0.50 (−0.96, −0.04)	−0.35 (−0.80, 0.09)	0.6548

**Table 5 nutrients-12-01883-t005:** Changes in bowel habits variables (depositions/day, stool consistency) between visits. T1: Initiation of first cereal, T2: Follow-up of first cereal, T3: Initiation of second cereal, T4: Follow-up of second cereal.

Outcome var.	Time	Group AB	Group BA	*p*-Value
**Depositions/day**	at T1	1.73 (1.39, 2.06)	1.40 (1.07, 1.72)	0.1682
	T2–T1	0.09 (−0.29, 0.48)	0.19 (−0.18, 0.56)	0.7225
	T3–T2	−0.05 (−0.43, 0.34)	0.09 (−0.27, 0.46)	0.6089
	T4–T3	−0.00 (−0.38, 0.38)	−0.07 (−0.44, 0.30)	0.7887
**Stool consistency**	at T1	2.38 (2.15, 2.61)	2.39 (2.17, 2.60)	0.9617
	T2–T1	0.31 (0.01, 0.62)	0.55 (0.26, 0.84)	0.2682
	T3–T2	0.02 (−0.28, 0.33)	−0.16 (−0.45, 0.13)	0.3940
	T4–T3	0.03 (−0.28, 0.34)	0.05 (−0.24, 0.34)	0.9199

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
