# Peer review of "Are Sugar-Reduced and Whole Grain Infant Cereals Sensorially Accepted at Weaning? A Randomized Controlled Cross-Over Trial"

_nutrients, 2020, doi:10.3390/nu12061883_

Round 1

Reviewer 1 Report

Drs Sanchez-Siles et al describe a crossover study on infant cereals in the weaning phase / introduction of solids from milk feeding.

The topic is relevant as the early life diet might have longterm impacts on metabolic outcomes.

Given the challenges of designing such a study (subjectivity of some of the parent-reported items) the current study design seems appropriate. 

The data are presented clearly. 

The discussion is comprehensive and should be shortened as should the reference section.

Author Response

Drs Sanchez-Siles et al. describe a crossover study on infant cereals in the weaning phase / introduction of solids from milk feeding. The topic is relevant as the early life diet might have long term impacts on metabolic outcomes. Given the challenges of designing such a study (subjectivity of some of the parent-reported items) the current study design seems appropriate. The data are presented clearly.

Response: Thank you for the very positive comments and constructive feedback. We do appreciate the time you have spent reviewing our manuscript and your careful read.

The discussion is comprehensive and should be shortened as should the reference section.

Response: Based on your comment, we have shortened mainly the part that describes the phytate content in whole grain cereals, which can be found in the discussion section. By summarising this part, we hope that we have found the right balance. Furthermore, we have achieved to reduce the reference section by removing 12 references.

Reviewer 2 Report

Undoubtedly the manuscript deal with a diet-health-related topic.

My main concern is on an ethical issue. Maybe this derives from the lack of description, so I invite the Authors to add an explanation in this regard.

The Authors report the study followed the Declaration of Helsinki (please add a reference) but I suggest explaining whether the Authors paid attention to some details inherent the protection of participants: 1) was an allergy test performed prior to the inclusion of subjects in the study? 2) was the nutritional composition of the administered foods compared to upper limits? 3) was the safety of each food product certified?

Overall, how the possibility to prevent adverse effects like those reported in the “3.3.4. Adverse events” paragraph?

Another question. I understand the interesting idea that it is possible to completely remove the added sugars from products. I feel this does not concern naturally occurring soluble carbohydrates. Am I right? Maybe, this could be explained a little bit.

  • Methods

How had been the participants contacted? The Authors say they were selected in healthcare centers but not how they were contacted. Moreover, would have been they randomly selected?

Who did collect the sample and the data?

Had been parents trained to evaluate in a standard way the child reactions?

  • Minor remarks

Lines 162-163 “sign of content” should be “signs of contentment”

Line 240: please report also here what the Authors have indicated in figure 1 (one subject has declined).

Author Response

We would like to thank you for your valuable and interesting comments. We do appreciate the time you have spent reviewing our manuscript and your careful read. We have considered all your comments and made all the necessary revisions, which we outline below.

Undoubtedly the manuscript deal with a diet-health-related topic. My main concern is on an ethical issue. Maybe this derives from the lack of description, so I invite the Authors to add an explanation in this regard. The Authors report the study followed the Declaration of Helsinki (please add a reference)

Response: Per your suggestion, in line 99-100 we have included the following: “The study was conducted according to the guidelines established in the Declaration of Helsinki (2013) [25]

We have also included this reference in the reference section:

[25]. World Medical Association. World Medical Association Declaration of Helsinki: Ethical Principles for Medical Research Involving Human Subjects. JAMA. 2013, 310, 2191–2194. doi:10.1001/jama.2013.281053

but I suggest explaining whether the Authors paid attention to some details inherent the protection of participants: 1) was an allergy test performed prior to the inclusion of subjects in the study?

Response: The inclusion of subjects in our study was carried out by pediatricians, and the exclusion criteria included children with allergy or intolerance to foods (this information can be found in line 97).

2) was the nutritional composition of the administered foods compared to upper limits? And 3) was the safety of each food product certified?

Response: Both infant cereals tested were already commercialized in the Spanish market, and its design and commercialization were in full compliance with all European Directives and Regulations.

Per your suggestion, in line 105-106 we have included the following: “The two infant cereals used in this experiment were in full compliance with European Directives and Regulations [26, 27], and commercially available products from Hero España S.A (Murcia, Spain)”.

We have also included these references in the reference section:

[26]. Commission Directive 2006/125/EC of 5 December 2006 on processed cereal-based foods and baby foods for infants and young children. OJ L 339, 6.12.2006, 16–35

[27]. Commission Regulation (EC) No 1881/2006 of 19 December 2006 setting maximum levels for certain contaminants in foodstuffs. OJ L 364, 20.12.2006, 5–24 and further modifications.

Overall, how the possibility to prevent adverse effects like those reported in the “3.3.4. Adverse events” paragraph?

Response: We did not find differences in adverse events between the infants fed with cereal A or cereal B. The adverse events reported were the typical issues that can occur in infants during complementary feeding, not related to specifically the consumption of infant cereals. E.g. upper respiratory tract infections including otitis, lower respiratory tract infections and gastrointestinal infections including parasitosis.

Another question. I understand the interesting idea that it is possible to completely remove the added sugars from products. I feel this does not concern naturally occurring soluble carbohydrates. Am I right? Maybe, this could be explained a little bit.

Response: Yes, you are right. The sugar in infant cereals can either be added, produced by starch hydrolyzation during processing, and naturally present in grains. The content of naturally present sugar can vary according to the type of cereals; however, in all cases, it is less than 3 g/100 g of cereals (USDA database).

Per your suggestion, in line 66-69 we have included the following: “The sugars in infant cereals are either added (mostly as sucrose), formed when the starch in the cereals is enzymatically hydrolyzed during processing (producing mainly glucose, maltose and isomaltose) [11], or naturally present. The sugar content naturally present in cereals range from 0 to 2.5 g/100 g, depending on the cereal type [23].”

We have also included this reference in the reference section:

[23]. U.S. Department of Agriculture (USDA), Agricultural Research Service. FoodData Central 2020. Available online: https://fdc.nal.usda.gov/ accessed on June 2020.

  • Methods

How had been the participants contacted? The Authors say they were selected in healthcare centers but not how they were contacted.

Response: The infants were assigned subjects in each healthcare center for primary care (in Spain the revision of healthy children by a pediatrician from infancy to 14 years of age is compulsory). Indeed, during the usual revision at 4 months of age and if the infant met the inclusion criteria, the pediatrician informed the parents about the study and invited those who met the inclusion criteria. If parents agreed to participate, written informed parental consent was obtained.

Moreover, would have been they randomly selected?

Response: After the first interview with the parent and if the subject met the inclusion criteria, the pediatricians invited them to participate in the study, and if the parent agreed on this, they signed the written informed parental consent. Then, the subjects were cited for visit 1 where the infants were randomly assigned to an intervention group (A or B).

Who did collect the sample and the data? Had been parents trained to evaluate in a standard way the child reactions?

Response:The data were collected with two types of questionnaires. Pediatricians’ questionnaires included infant characteristics, anthropometric measurements and adverse events. Parents’ questionnaires included the sensory test, variables of gastrointestinal tolerance, and stool characteristics” (p. 4 line 142)

Moreover, Pediatricians and parents were trained by qualified members of the research team on how to fill in the questionnaires and collect the data. Parents were also trained by qualified members of the research team on how to collect stool samples. The samples were frozen immediately after collection by the parents at home and stored until they were taken to a central collection site (healthcare center).

  • Minor remarks

Lines 162-163 “sign of content” should be “signs of contentment”

Response: you for your careful read. We apologize for the mistake. We have changed in line 166 the word “content” for “contentment”

Line 240: please report also here what the Authors have indicated in figure 1 (one subject has declined).

Response: your suggestion, in line 243 we have included the following: “A total of 49 infants were eligible for inclusion, one subject was declined and 48 were included in the study (Feeding sequence AB: n= 22; feeding sequence BA: n=26)”

Round 2

Reviewer 2 Report

The manuscript has been enhanced. However, I would furtherly ask for an additional explanation to reinforce ethical aspects.

The Author added the regulation about foods. That is good. However, again I would ask to report preventive actions to avoid pain for children.

Although a food is perfect from the formulation and processing point of view, allergy and other intolerance can cause sufferings. Children are vulnerable persons, so adults and professionals are obliged to prevent any possible adverse situations, so that adverse situations remain really accidentals and residuals.

24 adverse events are a huge number for 49 subjects. Which is the problem they could have had? Did some errors occur in evaluating the age to start the administration of cereal products?

Did the Authors test for allergy and celiac disease?

Moreover, in lines 159-160 “Adverse events were monitored throughout the study”  is written.

The Authors should also say which timely interventions were undertaken to limit at the minimum the pain for children.

Author Response

The manuscript has been enhanced. However, I would furtherly ask for an additional explanation to reinforce ethical aspects.

Response: Thanks again for your careful read and your constructive feedback. We do appreciate the time you have spent reviewing our manuscript. We have included in the manuscript additional explanations to reinforce ethical aspects, especially in the methodology section.

The Author added the regulation about foods. That is good. However, again I would ask to report preventive actions to avoid pain for children.

Although a food is perfect from the formulation and processing point of view, allergy and other intolerance can cause sufferings. Children are vulnerable persons, so adults and professionals are obliged to prevent any possible adverse situations, so that adverse situations remain really accidentals and residuals.

Response: We understand your concern as we are taking about a very sensitive population. Our research team has been conducted similar studies for a number of years. We are well aware of the measures that need to be considered while conducting these clinical trials. Maybe, we were not able to make them evident in our previous version of the manuscript. Still, our study was conducted according to all international guidelines (Good clinical practices and declaration of Helsinki) and was approved by the Ethics Committee of an Spanish National Health System Hospital (La Arrixaca, Murcia, Spain) and registered in ClinicalTrials.gov. In addition, it worth noting that in Spain, each infant is assigned to a specific paediatrician in a primary care healthcare center for periodic revision up to 14 years of age. Importantly, these paediatricians, who participate in our study, were the ones already assigned to infants in our study by the Spanish National Health System. Therefore, the children were monitored throughout their childhood by the same pediatrician who took care to prevent any possible adverse situations while our study was conducted. We now provide this information as follows:

  • Line 99-103 (Section 2.1 Subjects): “Pediatricians from the primary care health centers who collaborated in this study were the ones who were assigned by law to the infants of this study. Those pediatricians did all periodic revisions and monitored any possible adverse events (AEs). An AEs alert system was specially designed to detect not only the AEs usually related to complementary feeding (intolerances, allergies...), but any other sign or symptom of discomfort in children”
  • Line 204-213 (Section 2.4.2.4 Adverse events): “Although no AEs derived from a common and habitual practice was expected, any possible AE was monitored. Indeed, an AEs notification system, similar to that of a clinical trial with drugs, was designed following international recommendations [34]. The AEs alert system was specially designed to detect not only the AEs usually related to complementary feeding (intolerances, allergies ...) but any other sign or symptom of discomfort in children. All reported symptoms were considered AEs, even though they were not related to the intervention. AEs were classified as mild, moderate, and severe, and according to their possible relationship with complementary feeding as unrelated, possibly related, and related to the intervention. Any serious AEs or in possible relation to the intervention detected by the pediatrician had to be reported immediately to the principal investigator”.

24 adverse events are a huge number for 49 subjects. Which is the problem they could have had?

Response: Thank you very much for your comment. We would like to clarify that as this is a trial evaluating two previously recommended ways to feed a baby (cereal mainly based in refined flours vs. cereal enriched in whole grain), we did not expect any of our children to experience pain or adverse effects from the intervention. As expected, paediatricians did not report any adverse effects related to cereal intake or any other food. All adverse effects reported are extremely common among infants at this age due their immature immune system (upper and lower respiratory symptoms due to virus infections i.e., cough, bronchiolitis, otitis, common cold, etc….

This “high” number of reported side effects is precisely a consequence of the great effort made by pediatricians to report any symptoms that children may have had during the study period, whether or not they were related to the introduction of cereals.

The incidence of these mild, nonspecific clinical manifestations was similar in both groups, and similar to that of any other series of children of this age followed for 14 weeks (see for example Sierra C et al, Eur J Nutr. 2015; 54: 89–99; doi: 10.1007/s00394-014-0689-9)

We have also realized that the way some of the results are expressed in the manuscript could lead to a misinterpretation. The following sentence “In total, 24 mild adverse events (AEs) were reported among 17 children when taking Cereal A, and another 24 mild AEs were reported in 14 children while taking Cereal B” does not imply that 31 out of 46 children had AEs. This study concerns a cross-over trial and therefore the reported AEs were, for some cases, observed within the same subjects.

In other words, of a total of 46 subjects, no AEs were reported during the two study periods (7 + 7 weeks) for 22 subjects (48%). AEs were reported for 24 subjects. None of the AEs that occurred were related to the intake of the infant cereals or other foods. 77% of the AEs included respiratory infections (upper and lower tract), coinciding with the incorporation of the subjects to the kindergarten and the autumn season. Of 24 subjects that showed AEs, 13 subjects had one AE (of which 9 subjects had respiratory tract infections, including otitis), 8 subjects had between 2 and 3 AEs, and 3 subjects had between 4 and 6 AEs

To facilitate the interpretation of this matter, we have rewritten the section 3.3.4 Adverse events (lines 378-400)

Did some errors occur in evaluating the age to start the administration of cereal products?

Response: In the study protocol, we left it to the pediatrician’s discretion and decision to incorporate infant cereals into the children’s diet. In our study this happened at a mean age of 5.2±0.4 months. This mean age is in line with the European recommendations for complementary feeding (ESPGHAN). Regardless of this study, children would have started around this age with the complementary feeding period upon the pediatrician’s advice – specifically with infant cereals as these are one of the most common solids to start with in Spanish children.

To clarify this matter, we have incorporated a sentence in the methods section 2.2 Products (lines 113-115): ”Incorporation of infant cereals into the children’s diet was upon pediatrician’s recommendation, which in our study occurred at a mean age of 5.2±0.39 months. This is in line with the European recommendations for complementary feeding [1,8].”

Did the Authors test for allergy and celiac disease?

Response: We have not performed initial tests for allergies or celiac disease, as one of our exclusion criteria was “children with food allergies or intolerances”. The children included in our study were already “healthy subjects” of the collaborating pediatricians and they were monitored throughout the study, as explained earlier. The pediatricians did not include children in the study that were already familiar with food allergies or intolerances before. If at any point during the study the pediatricians detected any symptoms of an allergy or celiac disease, this child would have had their cereal intake stopped immediately and would have been excluded from the study. In any case, in our study there were no symptoms related to allergies or celiac disease observed.

Moreover, in lines 159-160 “Adverse events were monitored throughout the study” is written. The Authors should also say which timely interventions were undertaken to limit at the minimum the pain for children

Response: We hope we have clarified this issue in the new version of the manuscript. More concretely:

  • in the adverse events section, where we have added an explanation of the AEs notification system (lines 205-208) and the procedure that paediatricians followed to report any serious AEs to the principal investigator. (lines 211-213)
  • In the subject’s section (lines 99-103), where we have explained that throughout the whole study period the children were controlled for any issues by their pediatricians (the ones that are assigned by law in the Spanish National Healthcare System and participated in our study)

We would like to insist that as this is a trial evaluating two previously recommended ways to feed a baby, we did not expect any of our children to experience pain or adverse events from the intervention. Anyhow, for a better clarification we have also added the following text to the Results section (3.3.4 Adverse events) at lines 378-382: “The children in our study did not suffer any pain and did not find any difficulties with the incorporation of infant cereals in their diet. The infant cereals were introduced at an adequate age in line with the recommendations of the pediatricians and the European recommendations for complementary feeding. Throughout the whole study period the children were controlled for any issues by their pediatricians”

We again appreciate your thoughtful review.  We really tried to address every question and comment and we look forward to meeting your expectations.